Climate envelope predictions indicate an enlarged suitable wintering distribution for Great Bustards (Otis tarda dybowskii) in China for the 21st century

Mi Chunrong 1 2
Falk Huettmann 3
Guo Yumin 1 guoyumin@bjfu.edu.cn
1 College of Nature Conservation, Beijing Forestry University , Beijing , China
2 Key Laboratory of Water Cycle and Related Land Surface Processes, Institute of Geographic Sciences and Natural Resources Research, University of Chinese Academy of Sciences , Beijing , China
3 EWHALE Lab, Department of Biology and Wildlife, Institute of Arctic Biology, University of Alaska Fairbanks , AK , USA
Elphick Chris
Electronic publication date: 2016 Feb 1
Publication date: 2016
Volume: 4
Electronic Location ID: e1630
Received 2015 Jul 24; Accepted 2016 Jan 6
Copyright: © 2016 Mi et al.
Copyright year: 2016
Copyright holder: Mi et al.
License: This is an open access article distributed under the terms of the Creative Commons Attribution License, which permits unrestricted use, distribution, reproduction and adaptation in any medium and for any purpose provided that it is properly attributed. For attribution, the original author(s), title, publication source (PeerJ) and either DOI or URL of the article must be cited.
License URL: https://creativecommons.org/licenses/by/4.0/

Keywords: Climate change, Species distribution models (SDMs), Great Bustard (Otis tarda dybowskii), Random Forest, China

Funding: This research was funded by The State Forestry Administration of China. The funders had no role in study design, data collection and analysis, decision to publish, or preparation of the manuscript.

==============================
The rapidly changing climate makes humans realize that there is a critical need to incorporate climate change adaptation into conservation planning. Whether the wintering habitats of Great Bustards (Otis tarda dybowskii), a globally endangered migratory subspecies whose population is approximately 1,500–2,200 individuals in China, would be still suitable in a changing climate environment, and where this could be found, is an important protection issue. In this study, we selected the most suitable species distribution model for bustards using climate envelopes from four machine learning models, combining two modelling approaches (TreeNet and Random Forest) with two sets of variables (correlated variables removed or not). We used common evaluation methods area under the receiver operating characteristic curves (AUC) and the True Skill Statistic (TSS) as well as independent test data to identify the most suitable model. As often found elsewhere, we found Random Forest with all environmental variables outperformed in all assessment methods. When we projected the best model to the latest IPCC-CMIP5 climate scenarios (Representative Concentration Pathways (RCPs) 2.6, 4.5 and 8.5 in three Global Circulation Models (GCMs)), and averaged the project results of the three models, we found that suitable wintering habitats in the current bustard distribution would increase during the 21st century. The Northeast Plain and the south of North China were projected to become two major wintering areas for bustards. However, the models suggest that some currently suitable habitats will experience a reduction, such as Dongting Lake and Poyang Lake in the Middle and Lower Yangtze River Basin. Although our results suggested that suitable habitats in China would widen with climate change, greater efforts should be undertaken to assess and mitigate unstudied human disturbance, such as pollution, hunting, agricultural development, infrastructure construction, habitat fragmentation, and oil and mine exploitation. All of these are negatively and intensely linked with global change.

Introduction

Climate is among the most dominant factors that affect species across broad spatial scales (Woodward, 1987; Pearson & Dawson, 2003). Long-term studies indicate that the anomalous climate of the last half-century is already affecting the physiology, distribution, and phenology of many species, especially for many of the already endangered species (Sykes & Prentice, 1996; Hughes, 2000). Species distribution models (SDMs) are able to successfully quantify the relationship between species distribution and climate (Drew, Wiersma & Huettmann, 2011). Increasing attention has been given to projecting potential species distributions under various climate change scenarios by applying those methods (Dyer, 1995; Iverson & Prasad, 1998; Prasad, Iverson & Liaw, 2006; Wu et al., 2012), and incorporating climate change impacts into species conservation strategies (Araújo & Rahbek, 2006; Strange et al., 2011; Baltensperger & Huettmann, 2015).

Knowing species distributions represents an essential foundation in conservation biology (Araujo & Guisan, 2006; Tanneberger et al., 2010; Drew, Wiersma & Huettmann, 2011). Understanding where species emerge temporally and spatially across large geographic areas is important to conserving, monitoring, and managing species effectively (Wu & Smeins, 2000). For this purpose, SDMs, including process-based and bioclimatic envelope approaches, have been suggested as an effective tool (Guisan & Thuiller, 2005; Elith et al., 2006; Hu, Hu & Jiang, 2010). There has been rapid progress in this field of SDMs, and tools and workflows are now openly available to assess distributions and the impacts of climate change on species and habitats (Peterson et al., 2002; Hijmans & Graham, 2006; Drew, Wiersma & Huettmann, 2011).

The Great Bustard (Otis tarda) is one of the world’s heaviest flying bird and occupies grassland habitats. It is categorized as a globally vulnerable (VU) species according to the IUCN. Its world population in 2010 was estimated to be 44,100 to 57,000 individuals, and approximately 4–10% of the global population is located in China and believed to be declining (Alonso & Palacín, 2010). This species is divided into two subspecies: O. t. tarda and O. t. dybowskii. The latter subspecies (Taxonomic Serial No.:707876) is our research target. It is distributed throughout eastern Asia in areas such as Russia, Mongolia, China, and South and North Korea (Kong & Li, 2005). In China, O. t. dybowskii is distributed in Heilongjiang, Jilin, Inner Mongolia, and Hebei Province during summer. It winters in Heilongjiang, Jilin, Inner Mongolia, Shaanxi, Hebei, Henan, Shandong, Jiangsu, Jiangxi, Hubei, Hunan (Jiang, 2003; Wang & Yan, 2002), Shanxi, and Anhui Province (Wu & Liu, 2001). Until the early 20th century, there was a large population of O. t. dybowskii in Asia, with eastern Russia alone estimated to have held more than 50,000 individuals prior to the 1940s (Chan & Goroshko, 1998). However, numbers have declined during the 20th century, with a particularly rapid drop in counts from the wintering grounds during the 1950s and 1960s (Chan & Goroshko, 1998). Taking Poyang Lake, Jiangxi Province, China, as an example, hundreds of bustards were present in winter until the 1980s (Kennerley, 1987). By the late 1990s fewer than 20 individuals could be found (Wang, 1999), and in the last 10 years, bustards have not been observed. The wintering population of O. t. dybowskii in China was recently estimated at only 1,500–2,200 individuals (Goroshko, 2010). This rapid decline of the past four decades is seemingly linked to more efficient methods of hunting, the large-scale conversion of steppe to agricultural land on the breeding grounds, and habitat loss on the wintering grounds in China (Chan & Goroshko, 1998).

How to protect O. t. dybowskii and to keep this subspecies alive in the next 100 years, remains a non-trivial question to be resolved. In order to assess the likely effect of climate change on bustards in the 21st century, we employed species distribution models based on machine learning (TreeNet and Random Forest) to predict the distribution of habitats for this subspecies in the future. To the best of our knowledge, this work is the first predictive, spatial model of the wintering distribution of Great Bustards and it presents a step toward developing a national conservation effort to assess the management of bustards. At a minimum, the results of this study are expected to provide information on what habitat changes may occur, and guide future sampling, surveying, and conservation efforts across China. Further, we try to infer the wider status of this bird during times of global change.

Materials and Methods

Study area and data

The species data used in this study came from our own fieldwork investigations of 2012 and 2013, consisting of recorded bird occurrence GPS locations, and extracted data from published papers in Chinese journals, all of which we mapped in ArcGIS10.1 (see Supplement S1). Overall, we used 102 geo-referenced bird sighting locations across China from the period 1990 to 2013. Because of the lack of wintering data in Russia and Mongolia, we restricted our projected area just to China (Fig. 1). The boundaries of nature reserves were downloaded from the World Database on Protected Areas (WDPA, http://www.protectedplanet.net/) and clipped to the range of China in ArcGIS 10.1.

Figure 1 The study area for predicting the distribution of Great Bustards (Otis tarda dybowskii) in China.

102 presence records are shown; the elevation of this study area ranges from 0 to 8,233 m.

Nineteen bioclimatic variables were obtained from the WorldClim database (Hijmans et al., 2005, http://www.worldclim.org/) to describe for current climate conditions during 1950–2000. Other environmental variables that are considered to be important drivers of the Great Bustard’s distributions were also used to build the habitat distribution model. Those included topographical factors (altitude, slope, and aspect), water-related factors (distance to river, distance to lake, distance to coastline), human interference factors (distance to road, distance to rail road, and distance to settlement), and land cover. Aspect and slope layers were derived in ArcGIS 10.1 from the altitude layer obtained from the WorldClim database. Road, rail road, river, lake and coastline and settlement maps were taken from the Natural Earth database, while the land cover map, at 30s resolution and classified as 23 categories for the year of 2000, was taken from Global Land Cover 2000 database (detailed information is provided for all layers in Supplement S2). All spatial layers were resampled to a resolution of 30 seconds to correspond to that of the bioclimatic variables. Reliable future projections of land cover, distance to road, distance to rail road, distance to settlement, distance to river, and distance to lake predictors are not available. Including static variables based on current information in SDMs alongside dynamic variables could improve model performance (Stanton et al., 2012); therefore, we kept these variables in our future projections.

Machine learning models are difficult to overfit, especially Random Forest and methods that employ bagging (Breiman, 2001a). However, for a more conservative approach, we first calculated correlations among the 19 bioclimatic and 10 other environmental variables in ArcGIS and removed variables whenever a correlation coefficient >|0.90| was obtained (Costa et al., 2010; see correlation matrix in Supplement S3). A total of 15 bioclimate variables were removed, leaving 4 bioclimatic variables and 10 other environmental variables. Subsequently, we constructed two sets of bustard distribution models: one based on the reduced set of 14 predictors; the other used all 29 predictors. The models were named TN14, TN29, RF14 and RF29, where TN denotes a TreeNet analysis and RF denotes a Random Forest analysis.

Species distribution modeling and testing

We chose the TreeNet (generally referred to as Boosted Regression Trees (BRT), stochastic gradient boosting, Friedman, 2002) and Random Forest (Breiman, 2001a) algorithms produced by Salford Systems Ltd. to build our species distribution models because of their good performance and common usage (Zhai & Li, 2003; Elith et al., 2006; Drew, Wiersma & Huettmann, 2011; Lei et al., 2011). For more details on TreeNet and Random Forest, we refer readers to read the user guide (https://www.salford-systems.com/products/spm/userguide) and references within (see also Breiman, 2001b; Drew, Wiersma & Huettmann, 2011). Approximately 10,000 pseudo-absence points were taken by random sampling across all of China using the freely available Geospatial Modeling Environment software (Hawth’s Tools). We used a 10-fold cross-validation procedure for TN, where it divided our dataset 10-fold using 80% of the data for model calibration and retaining 20% of the data for evaluation, and out of bag data used to test RF. In addition, we used balanced class weights, and 1000 trees were built for all models to find an optimum within.

For model assessments, independent Great Bustard location records during 1980–2000 were acquired from the book of the Threatened Birds of Asia (Collar, Crosby & Crosby, 2001, see Supplement S4). We extracted the Relative Index of Occurrence (RIO) for these testing data from four projected maps (TN14, TN29, RF14, RF29). Boxplots with 95% confidence intervals for these RIO value were used to analyze the fit of each model. Furthermore, the testing and pseudo-absence points were used to calculate area under the receiver Operating Characteristic Curves (AUC) and the True Skill Statistic (TSS) (Allouche, Tsoar & Kadmon, 2006) using the ‘SDMTools’ package in R 3.1.0. The best suitable SDM for bustards was determined by comparing the boxplots, AUC and TSS of all models in concert.

Future projections for Great Bustards

After determining the final model technique, we constructed models for future climate scenarios for 2070 (average for 2061–2080). The data applied here are the most recent IPCC-CMIP5 climate projections from three Global Circulation Models (GCMs), BCC-CSM1-1, CNRM-CM5 and MIROC-ESM (hereafter BC, CN and MR, under three Representative Concentration Pathways (RCPs) 2.6, 4.5 and 8.5, which are named after a possible range of radiative forcing values in the year 2100 relative to pre-industrial values (+2.6, +4.5, +6.0, and +8.5 W/m2, respectively). We used the average predicted probability of occurrence across the three GCMs for each grid as our consensus forecast (named BCM). This method was considered as one of the best for developing an ensemble forecast (Hole et al., 2009). Subsequently, we applied the sensitivity-specificity equality approach as the suitable habitat threshold using a threshold probability of 0.85 to define the presence-absence distribution of Great Bustard wintering habitats as this has been found to be a robust approach (Liu et al., 2005).

Spatial analysis of potential effects of climate change envelopes

We used ArcGIS 10.1 to calculate the suitable habitat area of Great Bustards for two time periods (current and 2070) under three scenarios (RCP 2.6, 4.5 and 8.5) from three GCMs (BC, CN and MR) and their average (BCM). We also used the overlay analysis to assess the potential distribution changes of bustard wintering habitats, which allowed us to identify areas of the habitat range that are projected to be lost, gained or remain under future climate scenarios. Also, we overlaid four presence-absence distribution maps (current, RCP2.6, RCP4.5 and RCP8.5) with the boundaries of China’s nature reserves to explore how much Great Bustard habitat is currently found in reserves, and how that amount is projected to vary with climate change.

Results

Boxplots created using the independent test data taken from Collar, Crosby & Crosby (2001) indicated that the Random Forest model showed a higher relative index of occurrence (RIO) than the TreeNet model, and a stronger focus on a narrow range of values (>0.9; Fig. 2). The model based on 29 predictors performed a little better than the one based on 14 predictors, and was thus preferred for prediction.

Figure 2 Boxplots from independent test data taken from the literature (Collar, Crosby & Crosby, 2001) derived from four Great Bustards distribution models (TreeNet 14, TreeNet 29, Random Forest 14, Random Forest 29).

The high AUC values (>0.91) for all four Great Bustard models (Table 1) indicated that our models can accurately capture habitat relationships of bustards, as values above 0.75 generally indicate an adequate model performance for most applications (Pearce & Ferrier, 2000). AUCs of Random Forest models were higher than TreeNet models, and SDMs with 29 predictors performed better than the more parsimonious models with just 14 predictors. TSS had the same trends as AUC, and Random Forest performed better than TreeNet. Given these results, we selected a Random Forest model with 29 predictors as our final SDM with which to project future climate. The ranks of variable importance and one variable partial dependence plots for each predictor can be seen in Supplements S5 and S6.

Table 1 The AUC and TSS values of four Great Bustard distribution models.

Bold type indicates the best model according to each measure.

	TreeNet 14	TreeNet 29	Random Forest 14	Random Forest 29	
AUC	0.914	0.923	0.961	0.982	
TSS	0.828	0.846	0.922	0.965	

For an easier interpretation and assessment, we transformed four continues distribution maps (Current, BCM 2.6, BCM 4.5, BCM 8.5) with threshold of 0.85 to binary presence-absence maps. The results indicated that when solely judged by climate change envelopes the suitable wintering habitat of Great Bustards would enlarge (Fig. 3 and Table 2). Depending on RCPs of 2.6, 4.5 and 8.5 scenario, the suitable area was projected to increase between 4.7% and 28.8% to 2070 (Table 2).

Figure 3 Projected change of the Great Bustard’s suitable wintering habitat based on a consensus forecast (BCM) from three GCMs by 457 2070 under (A) RCP 2.6, (B) RCP 4.5, and (C) RCP 8.5.

The projected current distribution was overlaid with future projections to identify areas that would be lost, gained, or remain occupied.

Table 2 Projected change in the total area of the Great Bustard’s suitable winter habitat and the area in current nature reserve based on consensus forecast from three GCMs by 2070.

Areas are given in (km2), with the percent of the current total given in parentheses.

Scenario	Area lost (%)	Area remaining (%)	Area gained (%)	New total habitat	Habitat in reserve (%)	
Current	–	–	–	290,640	23,950 (8.2)	
RCP 2.6	67,290 (23.2)	223,350 (76.8)	218,360 (75.1)	374,410 (128.8)	29,530 (7.9)	
RCP 4.5	90,560 (31.2)	200,080 (68.8)	206,940 (71.2)	316,460 (108.9)	22,580 (7.1)	
RCP 8.5	88,080 (30.3)	202,560 (69.7)	189,690 (65.3)	304,170 (104.7)	27,300 (9.0)	

Table 2 and Fig. 3 show that 23 to 31% of the original suitable wintering habitats would be lost depending on RCP scenario (Table 2). Habitat would be severe near Dongting Lake, Poyang Lake in the Yangtze River Basin, and Tianjin, Beijing near Bohai Bay (see Fig. 3). Meanwhile, the long-term traditional wintering ground in Anhui, Jiangsu, Henan, Hebei, Shaanxi and Heilongjiang Provinces would remain. Our model shows that the area west of Shandong, the northeast of Henan, and the north of Jilin would gradually become suitable wintering grounds for Great Bustards (Fig. 3).

The expansion and shift of the habitats of bustards would also affect the conservation effectiveness of current reserves where this subspecies lives. Only about 8% (23,950 km2) of the current wintering habitat is located in nature reserves, but this area would increase under all three RCP scenarios (Table 2). Nonetheless, less than 10% of the bustard’s wintering distribution will be located in the nature reserve under all projections, and these reserves are mainly located in the west of Heilongjiang Province and the north of Jilin Province (Fig. 4).

Figure 4 Projection of Great Bustard habitat based on the consensus forecast from three GCMs overlaid with the locations of nature reserves.

(A) projected current distribution, (B) projected distribution by 2070 for RCP 2.6, (C) projected distribution by 2070 for RCP 4.5, (D) projected distribution by 2070 for RCP 8.5.

Discussion

Effective conservation of Great Bustards includes protection and restoration of their habitat. Our model is the first to predict and map, with high accuracy (AUC: 0.98, TSS: 0.94), the winter distribution of O. t. dybowskii in China. Our best climate envelope model was non-parsimonious (29 predictors) and based on the Random Forest algorithm, and indicates that suitable wintering habitats will increase during the 21st century (Table 2 and Fig. 3). However, some current habitat will become unsuitable, such as in the Dongting Lake and Poyang Lake areas in the Middle and Lower Yangtze River Basin. These are areas where observers have not seen any Great Bustards in the last ten years. Our forecast model showed that climate change also influenced population declines in both of the two lake regions (Fig. 3), except for efficient hunting and habitat loss because of human activity. In addition, we found that most wintering grounds (>90%) were not in nature reserves and carry no relevant area protection (see Table 2). Such findings are relevant for an improved understanding and prioritization of conservation efforts and suggest that new reserves should be established.

According to our model predictions, the Northeast Plain will become one of the major wintering areas for this subspecies. Originally, the Northeast Plain is the traditional breeding ground of bustards, and some male individuals overwinter there (Liu, 1997). Here we speculate that more bustards, both male and female individuals may remain there, and infer that this may result in bustards becoming resident in this area or having a shorter migration distance than currently. This situation has already been observed in the Red-crowned Crane (Grus japonensis) (Masatomi, Higashi & Masatomi, 2007) and with the Oriental White Stork (Ciconia boyciana) (Yang et al., 2007). The suitable wintering habitats in the Northeast Plain are located southeast of the Greater Khingan Mountains, southwest of the Lesser Khingan Mountains, as well as northwest of the Changbai Mountains. It is possible that these mountains might become a natural barrier to the habitat expansion of this subspecies. These areas are used for agriculture and are susceptible to urban expansion. Therefore, the question of how to leave enough space and how to protect and maintain this species under such a situation should be taken into serious consideration, before any new policies and conservation plans are made.

The southeast of Hebei and the northeast of Henan Province are the traditional wintering grounds for this species. There are at least 300 individuals overwintering in the commercially operated cropland of Cangzhou, Heibei Province and the Yellow River Wetland of Changyuan, Henan Province. However just a few loosely protected area exist in these areas.

To determine why so little bustard habitat is located in nature reserves, and which type of land cover bustards prefer during winter, we overlaid the presence-absence maps with a land cover layer, and quantified the land type of each grid cell of suitable habitat with ArcGIS 10.1. We found cropland and herbaceous cover were the bustards’ main wintering grounds in current and three future scenarios (Table 3; more detail is found in Supplement S7), with more than 74% of the wintering ground of this subspecies in cropland, a habitat not usually included in nature reserves. From these results, we can infer that this subspecies has become dependent on farmlands. This could be because no other habitats are left, and/or potentially because of the quantity of food left in farmland and the associated farmland planting mode. Other habitats, even within reserves, are not widely used. Suitable habitat environments such as flat terrain, and open landscapes with a far-reaching vision and with adequate food would help this endangered subspecies to overwinter and its population to increase again. Established seasonal protected areas, and also leaving more food behind on the wintering grounds may represent good management choices to protect bustards further in the landscape. Although the development of China and its landscapes is ongoing at a very fast pace, and urbanization is increasing, China is now paying more attention to the ecological role of nature and to environmental management. Furthermore, in order to ensure food security, we do not expect the area of cropland in China to decline in the 21st century and that the area of suitable wintering habitat will increase despite urbanization. From fieldwork and the reports of local villagers, we found that hunting (e.g. poisoned corn on farmland) was among the main factors killing bustards (e.g. Meng, 2010). Power lines are also a relevant threat (Raab & Schuetz, 2012).

Table 3 Land cover type for projections of Great Bustard wintering habitat under current conditions and three RCP projections by 2070.

Land type
scenario	Cropland (%)	Herbaceous cover (%)	Other (%)	Total	
Current	13,326 (74.7)	1,949 (10.9)	2,564 (14.4)	17,839	
RCP 2.6	20,133 (74.9)	3,758 (14.0)	2,998 (11.1)	26,889	
RCP 4.5	18,148 (74.3)	3,795 (15.5)	2,491 (10.2)	24,434	
RCP 8.5	17,228 (73.5)	3,782 (16.1)	2,446 (10.4)	23,456	

There is much that is unknown about Great Bustards in Asia. Knowledge of location and population data are also insufficient. We believe that more work of this kind should be undertaken in the future, including efforts to better address its status as an international migratory species. The most important work might consist of monitoring this subspecies in order to obtain fundamental data for effective conservation action.

Our distribution modelling is meant to better indicate where bustards stay during winter and to be applied for the management of this species. Based on our research finding, we are optimistic about the bustard’s wintering habitats. However, the breeding grounds located in the steppe land are severely affected by human activities, which has already resulted in massive habitat loss and habitat fragmentation. More research is urgently needed in the breeding grounds, including establishment of a monitoring network, a comprehensive distribution and abundance survey, and modelling of breeding distributions based on existing data. Finally, a suitable and effective plan is needed to protect this endangered species nationally and internationally.

The limitations of our research include: 1) our model is based on presence-pseudo absence data and we could not estimate current or future population size; 3) we lack future road, residential, and land cover scenarios, although such GIS layers would be of great value to conservation planning and would likely improved future projections; and 4) bustards also winter in Mongolia and Russia; our research is currently restricted to China because of we lack data from these other countries. We hope this research could help to trigger the collection of new information on those topics.

In summary, our results indicate that there is a critical need to incorporate climate change adaptation into our conservation planning during an already rapidly changing climate. Our model could aid managers currently addressing conservation of bustards in China and elsewhere. Distribution maps could be overlaid with maps of the current and predicted locations of activities such as oil, gas, mineral, and wind energy development, in order to identify areas of potential future conflict, estimate the potential severity of impacts caused by development, and prioritize conservation strategies geographically (cf. Beiring, 2014).

Supplemental Information

Supplemental Information 1 presence data.

Supplement S1 presence data.

Click here for additional data file.

Supplemental Information 2 Supplement S2 predictors information.

Supplement S2 predictors information.

Click here for additional data file.

Supplemental Information 3 correlation matrix.

Supplement S3 Correlation matrix.

Click here for additional data file.

Supplemental Information 4 Testing data.

Supplement S4 Test data.

Click here for additional data file.

Supplemental Information 5 Supplement S5 Ranks of variable importance.

Click here for additional data file.

Supplemental Information 6 Supplement S6 Partial dependence plots of each variable.

Click here for additional data file.

Supplemental Information 7 Supplement S7 Land cover type of suitable habitat.

Click here for additional data file.

This research was possible because of the large investment of field effort, money, time, personal interest, and dedication by researchers for the past 24+ years. We thank all those who contributed to the International Great Bustard Census, as well as students from the EWHALE lab, UAF and Salford Systems Ltd., as well as the China Great Bustard Protection and Monitoring Network (http://www.otistarda.org/en).

Additional Information and Declarations

Competing Interests

Author Contributions

Data Deposition

The authors declare that they have no competing interests.

Chunrong Mi conceived and designed the experiments, performed the experiments, analyzed the data, contributed reagents/materials/analysis tools, wrote the paper, prepared figures and/or tables, reviewed drafts of the paper.

Huettmann Falk analyzed the data, contributed reagents/materials/analysis tools, reviewed drafts of the paper.

Yumin Guo reviewed drafts of the paper.

The following information was supplied regarding data availability:

The research in this article did not generate any raw data.

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
