# Peer review of "Climate envelope predictions indicate an enlarged suitable wintering distribution for Great Bustards (Otis tarda dybowskii) in China for the 21st century"

_PeerJ, doi:10.7717/peerj.1630_

## Round 0.1 · original submission · Major Revisions

First, I apologize for how long this review has taken. We have been waiting for a third review, but unfortunately it has been delayed. Consequently, I have decided to move ahead with only two reviews and conduct a more thorough reading of my own.

The reviewers are positive about the general aims of the study, but have substantial comments about the current manuscript and a number of recommendations about how it could be improved. I agree with their concerns and would like to see each point carefully addressed should you choose to submit a revision. Of most concern are: (1) the issues related to correlated independent variables and overfitting (e.g., perhaps I missed it – and I only have a basic understanding of these methods – but I did not see a discussion of how the tuning parameter was optimized in the RF analyses, e.g., see the discussion here http://stats.stackexchange.com/questions/111968/random-forest-how-to-handle-overfitting); (2) the suggestion to add data from outside China to increase sample size and improve fundamental niche description; (3) the lack of information on how the presence data were obtained; and (4) the need for a clearer discussion of how this information can be used in conservation. This last point is especially relevant given stated goals of the paper and the caveats raised in the Discussion; in particular it would be useful to add an analysis, or at least a more detailed discussion, of how land use is likely to change in areas that bustards are projected to move into. I also cannot see how this model could be used to forecast population changes as claimed in the last paragraph (at least not without much more information on demographic processes, land use change, and so on).

I also feel that the current paper is overly long, given the knowledge gains that it currently contains, and could be shortened in various places. E.g., the lengthy paragraph on RT and BRT methods on p 16 seems largely unnecessary as these methods are becoming much more widely known. Moreover, if that paragraph were to be included, I would like to see it balanced with discussion of some of the drawbacks of machine learning methods (see below). Overall, I think – assuming no additional analyses are added – the paper could be shortened by ~30%.

The paper also warrants a very careful review for spelling, grammar, syntax, punctuation, etc. as I noted many errors in my review. I have not made detailed comments on these points as I anticipate that there will be substantial text changes in response to the reviewers’ comments, but would hope to see fewer problems in a revision.

Finally, a couple of additional specific comments from my review:

- It would be helpful to describe the different climate scenarios, rather than just using the IPCC labels –readers will not necessary know what the different RCPs are, yet they are central to the analysis.

- The statement that TN and RF are stand-alone software products is unclear to me – these techniques are statistical methods, not simply pieces of software, and can be done in various statistical packages (e.g. package randomforest in R). Is there a specific reason why this one piece of software is singled out in this way?

- The approach taken to assign habitat suitability thresholds should provide more detail. What is it about your knowledge of current distributions that you used? Also I am unclear what distinguishes “marginally”, “moderately”, and “highly” suitable. It is not clear to me how someone could replicate this portion of the study, or evaluate your judgments (e.g., in deciding whether to protect moderately vs highly suitable areas).

- Some discussion of the mechanisms behind the projected range shift would be useful. Without that ecological understanding it is difficult to know how uncertainties/changes in our understanding of climate change might alter the conclusions, or how land use changes might interfere with them. (Note that in my experience, although machine learning methods can be good for prediction or pattern detection, the difficulty of interpreting mechanisms can be a drawback compared to some other methods.) This point also relates to the comments on parsimony towards the end of the Discussion. I suspect that many would consider parsimony to be valuable because it allows for better mechanistic understanding, even if marginally better predictions can be obtained by including a lot more variables. This is how I have always interpreted Burnham and Anderson’s arguments (at least in part). There is certainly a debate to be had about how one should weight predictive power vs. understanding of mechanism – but I think that debate is more complex than the current manuscript suggests.

Reviewer 1 ·

Basic reporting

Results and raw data have been made sufficiently available and it is significant for a good understanding of the study. However, to improve the clarity of the manuscript, I have some suggestions for a good balance between main text/figures and Supplementary materials, and figures legends are very incomplete for a good understanding:
Figure 1: Repeat the Latin name of the studied species in the legend.
Figure 2: Explain specific names such as isectpntrst, GME, SPM…in the legend.
Figure 3: Repeat the name of the 4 models in the legend and the reference for “Threaten Birds of Asia” (idem for Table 1).
Table 3: The table is quite difficult to read, I think it is possible to illustrate the results with two barplots, 2050 and 2070, for each one of the 4 current suitability classes, providing the percentages of each classes. It could be done only for one scenario, maybe the intermediate one or the more severe, and the table could be moved to Supplementary material.

Minor comments:
I have noted a lot of typographical errors (for instance P2 L1) and some informal sentences (e.g. P6 L1, P17 L2-3…).
Standardize lower and upper case letters for Great Bustards throughout.
Prefer “projected” to “predicted” distributions throughout.

Experimental design

I have major concerns about the way to address the issue of potential overfitting of models. You compared the predictive accuracy of models with two sets of 29 and 14 predictive variables. Both sets have a very high number of variables, still highly correlated each which others.
I don’t think that a measure of prediction accuracy can show the overfitting of models. Although more complicated models may appear to give a better fit, the predictions they produce may be poorer (Chatfield, 1995).
Furthermore, conclusions are specific to the modelling techniques you used and cannot be generalized to all modelling techniques. While both methods, BRT and RF, are known to avoid overfitting (Elith et al. 2006, Breiman 2001), some others need careful attention about correlation of predictive variables. P17 L1-7 need to be carefully revised.

Different outputs of climate models (GCMs) can also lead to uncertainty in projections (Buisson et al. 2009, GCB). Why not taking it into account by using different GCMs, as well as RCPs, and/or using averaging methods to keep the central tendency given by the different GCMs?


Line-by-line comments:

P7 11 What do you mean by “land-cover”? Which unit or categories?

P7 20-21 unclear, you mean under the assumption of an increasing development of human activities?

P7 17-18 unclear.

P10 3-5 Methods and data source for the model evaluation with independent data should be more detailed and described above, with the other evaluation methods (cross-validation).
However, I found that the description of methods and (consistent) results for 5 evaluation method conduct to a long and quite difficult text. Maybe you could retain one method of evaluation within the five in the main text and describe the others and their results in Supplementary materials.

P10 5-10 I don’t understand how, from your comparison (Supplement S5), you select the sensitivity-specificity equality approach as the most suitable threshold.

P10 L17 What do you mean? Why did you select a threshold of presence/absence from a comparison of three methods if finally you arbitrarily choose thresholds?

P11 L7 How? by using the centroids of distributions or extreme coordinates? The results are not clear either (Table 4): in the legend I don’t understand what you mean with “distribution range” with 4 coordinates. It may be illustrated with a graph and the table be moved to Supplementary material.

P14 L1 Do you mean longitudinal and latitudinal?

P16 L7-18 Points iii) and iv) are particularly unclear. L17: what is it usually time-consuming in the preparation of data?

Validity of the findings

I have some major concerns concerning data used to project wintering range. While China represents a large part of the wintering range of the Great Bustard, it would be great to integrate the whole range with the few missing areas, not only to obtain a comprehensive study to join Northeast Asia conservation effort, but mainly because it is critical to use presence data from the largest geographical extent through the whole species range to better approximate the fundamental niche of the species from species distribution models (Araújo and Guisan, 2006; Phillips et al., 2006). Thereby, a study area that encompassed the whole wintering range of the species would be more ecologically meaningful than administrative boundaries. I am not sure that your results about eastward range shift were consistent when you integrate the closed region in Mongolia in study extent. Would it be possible to extend data from literature to bordering countries? At least, we miss a brief description of the known breeding and wintering range of this subspecies to better understand the location of your study area.

Furthermore, we miss some information about raw data.
How did you distinguish wintering from breeding occurrences? I guess that data in the frontiers with Mongolia could have been quite ambiguous?
P6 16-18 How occurrences data are obtained? From protocol (transects…) or occasional observations? Could we expect some difference in detection probability between male and female?

Additional comments

Here are some line-by-line comments about unclear parts of the text:

P2 L5 “survive” seems not be the most appropriate term.

P5 L3-5, add a reference if you do not further developed this assumptions, they are reviewed for instance by Guisan & Thuiller 2005 (Ecology Letters).

P6 L3 a reference for “early time”?

You give some insights for future work (P15 1-2 and P17 10-12) but I do not understand what you suggest.

P15 13 Why residential? is it a known location for breeding?

Reviewer 2 ·

Basic reporting

see General Comments to the Author

Experimental design

see General Comments to the Author

Validity of the findings

see General Comments to the Author

Additional comments

I appreciate the effort done in modeling, and the difficulties of gathering observations of this rare species in China. However, a lot of emphasis is put on strictly modeling methodological aspects, and it is not clear to me how this translates into clear, practical conclusions for the conservation of the species.

My main comments are:
1. The samples size is small and doesn't probably correspond to the requirements of the modeling techniques applied.
2. The ms is too long compared to the value of the results and conclusions, and should be significantly reduced, by concentrating on the main points.
3. Although the authors conclude that suitable wintering habitats in the current bustards distribution would increase during 19 the 21st century, they do not suggest how this conclusion translates into conservation or management actions that would help guaranteeing the species' survival.

I recommend:
The number of predictors should be reduced, because of the small sample size.
One way of doing this is to use a smaller number of predictors that have been shown to be relevant to great bustards in previous studies. Another way is to reduce the number of approaches. For example, I would suggest omitting the second approach used (page 8, lines 6-7), since it produces model overfitting through the use of highly correlated predictors.
The length of the manuscript can be reduced by deleting unnecessary theoretical discussions on global warming effects, or methodological aspects that do not directly affect this study. For example: reduce the first two paragraphs of the Introduction to just the main ideas and references; reduce unnecessary explanations in p 8 line 9 to p 9, line 5, and parts of p 9; reduce unnecessary details in Results and Discussion; include Fig 2, Table 1, Table 3 in Supplementary material.

---

## Round 0.2 · Minor Revisions

This is a nice revision and a greatly improved manuscript. The reviewer has a few follow up comments that need to be addressed and I have done a detailed edit to improve grammar and clarity, and to reduce redundancy (I will send the edited word document separately as the PeerJ system appears to only allow my to upload pdfs). In my edits I have tried hard not to alter meaning, but please check carefully to ensure I have not done so. I've also made a number of minor comments on the edited document (see margin balloons), some of which require action.

I should clarify one point that I made in my previous comments, as you appear to have misunderstood my meaning. When I suggested that machine learning methods are better for describing pattern than mechanisms, I was not suggesting that nothing can be interpreted from the results, only that mechanistic understanding is often limited. I agree with your inferences about shifts in projected distributions. But, the manuscript does not provide any mechanistic explanation for the projected shifts, other than that it is somehow related to the large collection of variables used (i.e., it does not explain precisely which aspects of climate will affect which habitats, and why those will affect the species distribution). In my experience, information of that type generally does not emerge from machine learning analyses. Thus, although the study does make clear predictions about what might happen in the future, it does not explain why. That is perfectly fine when the goal is only prediction (as it is here), but it remains a large limitation when devising conservation strategies. To take one example, this limitation would be a problem when there is uncertainty in variables used for projection (as there is for climate variables). E.g., if one knew that the projected shift was largely influenced by climate variables for which future scenarios are highly uncertainty, then one would view the projections more skeptically than if one knew that projected shifts were more influenced by variables that are much more certain. With the current method that type of assessment is impossible.

Reviewer 1 ·

Basic reporting

No comments

Experimental design

No comments

Validity of the findings

No comments

Additional comments

The authors addressed my comments and queries and the manuscript has been much improved from the earlier version. While the results & methods sections are now clear and well-written, some parts of the discussion would need to be revised. Here are my comments on the current version of the manuscript:

L123 Thanks to the authors for explaining me that “Land cover means land cover”. I consider that “Land cover” variable might require more relevant information (among resolution, classes of land cover, year, accuracy, reference…) than only one general link for a website, and that it would be also useful to precise that land cover is used as a categorical variable in the model.

L281 I think the overlay of presence/absence from models with land-cover can lead to potential circularity problem if the land-cover layer used for this analysis has been also used as predictors to model presence/absence. At least I think that land cover data could be overlay with true presence observations but not with the modelled presences. Anyway, the methods & results of this analysis presented in the discussion section should be presented earlier in the manuscript.

L298 Because actually we don’t know how land cover can evolve, I think this point should be nuanced and more discussed. The habitat shift of bustards from natural grassland to cropland, the rapid change, or the intensification of agricultural practices, might become ecological traps for bustards (Battin 2004, Cons. Bio.). For other species of bustards for instance, the intensification of agricultural practices has been identify as the cause of a severe decline of populations of Little Bustards in France (Bretagnolle et al. 2011, Ibis) and potential change represents a serious threat also for wintering populations of Houbara Bustards, for instance in Iran (Aghainajafi-Zadeh et al. 2010, J. Arid. Environ.).

L203-206 In this part, it is unclear if the authors compare their model parameterization with the one used within all available modeling techniques to model species distribution (not so meaningful I guess because the parameterization and the way to deal with overfitting is inherent to the modeling technique used) or if they compare with studies that used similar modeling techniques. Please precise and why not add some references for a more meaningful comparison.

Avoid using brackets to comment the results (L203-206, L244)

L320 Why do the authors consider that they have “somewhat undersampled” presences? Some areas of presence in China would be missing in the study? Explain better, or remove. 102 records are actually quite strong but if a sampling bias occurred within China, it would need to be more discussed here.

Besides, the focus on a reduced spatial extent (China) might be more discussed. Changes in other countries might have consequences on the future distribution of the subspecies also in China (changes in migration, migration distances, fate of breeding population…).


Line-by-line comments:

L25-28 The issue is not very well addressed here, I would prefer “whether the wintering habitats of Great Bustards … would be still suitable in China in a changing environment, and where, …”
L93 “assess more specifically the potential impact of” rather that “assess more specific the effect”
L130 Random Forest instead of randomForest
L138-139 That sentence arrives too early, you didn’t present what TN and RF mean yet.
L174 GCMs instead of GSMs
L487 “fourGreat”
L497 Precise in the legend what the text in bold means.
L255 results showed only a correlation with climate change and population declines, not a cause-effect relationship
L264-265 would instead of will
L276 & 278 If possible, adding references for these two sentences could guide the reader to find the original information elsewhere in the literature or reports.
L292-293 Add references for these not so “well-known management choices”
L319 Maybe the following paragraph could be shortened and that sentence could be directly linked with perspectives L326-329
L530 Table 3 Explain better as well as in Table 2 and add units.

---

## Round 0.3 · accepted · Accept

The new version does a nice job of addressing my previous comments. I have a few additional, minor editorial comments; I will send an annotated MS Word file in a separate email. Assuming I have not introduced errors, most changes can be incorporated simply by using the "Accept all" function. There is one place where a section of methods and results is embedded within the discussion - that information should be incorporated into the appropriate sections.